# Oral Treatment of Central Serous Chorioretinopathy Patients Using Propranolol Tablets

**DOI:** 10.3390/ph13110336

**Published:** 2020-10-23

**Authors:** Li-Chai Chen, Jui-Wen Ma, Po-Chuen Shieh, Chi-Ting Horng

**Affiliations:** 1Department of Pharmacy, Tajen University, Pingtung 907, Taiwan; icupdrab@tajen.edu.tw (L.-C.C.); a26154295@gmail.com (J.-W.M.); pochuen@tajen.edu.tw (P.-C.S.); 2Koahsiung Armed Forces General Hospital, Koahsiung 802, Taiwan

**Keywords:** central serous chorioretinopathy (CSCR), propranolol, best-corrected visual acuity (BCVA)

## Abstract

Purpose: To evaluate the pharmacological effects of propranolol treatment of patients with central serous chorioretinopathy (CSCR) over 4 months. Results: Among the 89 male and 31 female patients, the mean BCVA decreased to 0.42 ± 0.08 logMAR during CSCR attacks. Oral propranolol showed good effectiveness in reducing CSCR signs after at least 4 months of treatment. The final BCVA of the patients in groups 1 and 2 was 0.09 ± 0.01 and 0.19 ± 0.03 logMAR, respectively (*p* < 0.05). Moreover, the mean complete remission time in groups 1 and 2 was 1.9 and 3.5 months, respectively (*p* < 0.05), while the “success” rate in groups 1 and 2 was 95.0% (57/60) and 78.3% (47/60), respectively (*p* < 0.05). The recurrence rate in groups 1 and 2 was 5.3% (3/57) and 25.5% (12/47) after a further 5 months of follow-up, respectively (*p* < 0.05). Materials and Methods: One hundred and twenty patients were enrolled and randomly divided into two groups that both underwent a visual acuity test and optical coherence tomography (OCT) scanning, between April and December 2017. The 60 patients in group 1 were requested to take propranolol for 4 months, while the other 60 subjects (group 2) received placebo therapy during the same period. The best-corrected visual acuity (BCVA) of every volunteer and an OCT image of each patient were checked and recorded at the beginning of the study and each week thereafter. If the signs of CSCR disappeared completely from the OCT scans, the case was considered a “success” and treatment stopped at once. However, the “success” subjects were further evaluated in follow-ups throughout the next 5 months to determine the rate of recurrence in groups 1 and 2. The time of total complete remission of CSCR from the OCT scans was also measured in groups 1 and 2. Conclusion: CSCR patients revealed an excellent prognosis and success rate of 95.0% after taking propranolol. The treatment was able to enhance subretinal fluid (SRF) absorption, shorten the time to total complete remission, and significantly decrease CSCR recurrence. As such, we suggest that taking propranolol may be an alternative and viable choice for CSCR patients, given that the new method was shown to be safe, cheap, effective, well tolerated and convenient.

## 1. Introduction

Central serous chorioretinopathy (CSCR) is a type of macular disease that is characterized by serious retinal detachment, with retinal pigment epithelium (RPE) detachment and choroidal hyperpermeability. The main pathophysiology of CSCR is the accumulation of subretinal fluid (SRF) in the posterior pole. The exact pathophysiology of CSCR is its choroid circulatory abnormalities and RPE disturbance, resulting in SRF leaking from the impaired tight junction [1]. Hyperpermeability of the choroid can be caused by blood stasis, ischemia, or inflammation [2]. Although CSCR is considered a benign and self-limiting disease, it has a tendency to recur, resulting in SRF absorption for some patients. While most CSCR patients show spontaneous visual recovery at an average of four months, nearly one-third of patients suffer from significant visual acuity impairment [3]. Epidemiologically considered, CSCR primarily affects middle-aged men between 45 and 55 years old. Although in the majority of cases, CSCR can disappear spontaneously, a few patients suffer relapses of varying degrees. Recurrent CSCR has been reported in 40–50% of cases, which gradually develops into a chronic case with RPE atrophy and pigmentation in the macula [4]. For example, older patients are more likely to present with a diffuse loss of RPE, cystoid macular edema (CME), irregular serous macular detachment, secondary choroidal neovascularization (CNV) formation, and long-standing intraretinal cystoid cavities, which may lead to blindness [1].

In the acute presentation of CSCR, patients complain about various symptoms such as mild decreased vision, metamorphopsia, micropsia, and central scotoma due to the accumulation of SRF. In addition, macula with well-demarcated round areas and a loss of foveal reflex can be observed, which means that an increase in central macular thickness could provide a good hint toward diagnosis [5]. Additionally, some ocular parameters are also significantly decreased after treatment; for instance, central subfield thickness and subretinal fluid volume [6]. Moreover, there are many risk factors for CSCR, including male gender, mental stress, type A personality, hypertension, the use of steroids or psychotropic medication, preeclampsia in pregnancy, and some systemic disorders. In some rare cases, Cushing’s syndrome or some steroid-producing tumors also occur [1,7]. In clinics, up to 50% of patients typically present recurrence within the first year of initial occurrence. Furthermore, a few patients develop irreversible visual loss due to RPE atrophy, subretinal fibrosis, and CNV formation. Even though CSCR patients have good best-corrected visual acuity (BCVA), a lower density of retinal cone cells can be found, especially in the advanced stages of chronic CSCR with residual symptoms, including metamorphopsia, color deficiency, and decreased contrast sensitivity [8,9,10]. Severe CSCR sometimes results in the death of photoreceptors as well as retinal detachment (RD) in the chronic stage, which subsequently decrease visual acuity.

Most medical reports suggest that without aggressive treatment, RPE atrophy and CNV can worsen in cases of CSCR [11,12,13]. Regardless of the characteristics of spontaneous remission in CSCR, the trend is to treat CSCR rather than wait for possible recovery by itself. Therefore, early prevention, diagnosis, and treatment are necessary, which could possibly enable complete remission of SRF and rapid recovery of vision within the shortest possible time [14]. Therefore, blocking the progression of CSCR and enhancing the absorption of SRF have quickly become mainstays in CSCR treatment. Thus far, there are several treatments that have been developed for CSCR patients. For example, several years ago, steroids were initially administered to CSCR patients either orally or intravenously [7]. Since then, various therapies have been developed, including the popular approach of intravitreous injection (IVI) of bevacizumab (Avastin^®^, Roche, Switzerland) [15]. Moreover, Gramajo et al. reported that melatonin (3 g three times daily) may decrease the central thickness of macula and improve BCVA in chronic CSCR [16]. However, an aggressive approach has also been adopted for CSCR patients and includes thermal laser photocoagulation and photodynamic therapy, which may carry inherent risks and show poor results [17,18]. In this research, we attempted to develop a new method for treating patients with CSCR based on only taking propranolol. Beta-blockade is useful in clinical conditions in which sympathetic activities are excessive—in this case, the pathophysiology of CSCR—which may impact on visual function [19]. Furthermore, anxiety that increases the catecholamine level may also be a factor impacting CSCR. Therefore, we supposed that the β-blockers, such as the propanol that was used in this study, are potential drugs for curing CSCR [20].

## 2. Results

All 120 patients completed nine months of the study without significant complications or personal discomfort (e.g., chest discomfort or shortness of breath). All had good patient compliance to propranolol. Among the volunteers, 62 (51.6%) had CSCR in the right eye, and 58 in the left eye (48.4%); therefore, the incidence of CSCR for each eye was nearly equal. CSCR was primarily found in middle-aged men between 45 and 55 years of age [21]. In our study, the mean age of the 120 subjects was 45.5 ± 2.5 years, which is consistent with most other reports, and the majority were male (92; 76.7%), while there were only 28 females (23.3%) [22]. At the end of the third month, there were 104 “success” patients from across the two groups, with a total success rate of 86.6%. Most of the patients in group 1 revealed an improvement in BCVA and the complete disappearance of the dome shape from the OCT scans (Table 1). The success rate was approximately 95%, with only three failed cases (5.0%). In group 2, the success rate was 78.3% (47/60) using the placebo treatment. Therefore, patients showed significantly better outcomes after propranolol treatment when compared to the placebo therapy (*p* < 0.05). Furthermore, the mean BCVA of all 120 patients decreased to 0.42 ± 0.08 logMAR with CSCR compared to their first visit. After effective treatment, the final mean BCVA in group 1 validly increased to 0.09 ± 0.01 logMAR; however, the mean BCVA was only 0.19 ± 0.03 logMAR in group 2 (*p* < 0.05 *). As for the complete remission time, group 1 required a mean time of only 1.9 months to recover. However, for group 2, a longer time to remission (approximately 3.5 months) was needed (*p* < 0.05 *). The flow chart of the designed protocols and the final results of our study is shown in Figure 1. We compared the results between groups 1 and 2 using a chi-square test. A *p*-value < 0.05 was considered statistically significant and is marked with an asterisk.

## 3. Discussion

Central serous chorioretinopathy (CSCR) usually affects those between 20 and 50 years of age who exhibit acute or sub-acute central vision loss or distortion in daily life. Other chief complaints also include metamorphopsia, hyperopic or myopic shift, micropsia, central scotoma, and even decreased contrast sensitivity and color saturation [1]. When only one eye is involved, the patient also suffers from the loss of static and dynamic stereopsis and sometimes become disabled in their daily activities. Out of CSCR patients, 30% have bilateral involvement and 40% experience recurrence, while decreased visual acuity occurs with and/or results in persistent poor vision in 5–10% of chronic CSCR patients [7]. Males are nearly six times more likely to be affected than females [23]. In our study, the results also show a similar pattern, with 76.6% of our patients being male. In recent years, it has been suggested that male labor is the major source of family support worldwide. It has been proposed that the male population suffers from greater psychological stress and physiologic fatigue, which are etiologies of CSCR [24]. In addition, as many articles have revealed, the mean age of CSCR onset is approximately 40 years old, which is typically a peak point in an individuals’ economic responsibilities and work commitments in their daily lives. The mean age of the 120 patients in our study was 45.5 ± 2.5 years, which supports the idea that people in their 40s may be in the higher incidence group because they are responsible for the economic support of their families.

Although it is usually self-limiting, CSCR may recur chronically with permanent visual acuity deficits. Around 80–90% of CSCR patients in the acute stage are expected to show spontaneous visual recovery after approximately 4–6 months without medication [25]. In our study, the mean remission time in the placebo group (without drug treatment) was approximately 3.5 months, which is consistent with previous findings. In our design, oral propranolol (Inderal^®^, AstraZeneca, UK) tablets were prescribed by the same ophthalmologist (Dr. Horng) and used to treat the patients with CSCR. The results show that oral propranolol tablets shortened the disease course to a mean time of 1.9 months, and helped patients to promptly recover their BCVA. Furthermore, higher recurrence rates could happen even if the patients had received treatment. For example, following spironolactone treatment, a higher recurrence rate of approximately 48% was noted within 3–6 months after recovery [24]. Moreover, a 24.5% recurrence rate has also been observed after photodynamic therapy in subjects with chronic CSCR [26]. In our propranolol treatment, the recurrence rate was decreased to 5.3%, which is beneficial for CSCR patients.

For example, the use of ketoconazole, which inhibits steps in the synthesis of glucocorticoid, led to the complete diminishing of the dome shape in OCT scans (Table 2). The success rate in group 1 was approximately 95%, with only three failed cases (5.0%). In group 2, the success rate was 78.3% (47/60) with the placebo treatment. Therefore, patients showed significantly better outcomes for propranolol treatment compared to the placebo therapy (*p* < 0.05). Furthermore, the mean BCVA of all 120 patients decreased to 0.42 ± 0.08 logMAR with CSCR compared to the first visit. After effective treatment, the final mean BCVA in group 1 validly increased to 0.09 ± 0.01 logMAR; however, the mean BCVA was only 0.19 ± 0.03 logMAR in group 2 (*p* < 0.05). As for the complete remission time, it only took a mean time of 1.9 months to recover in group 1; however, more time (approximately 3.5 months) was needed in group 2 (*p* < 0.05).

CSCR is a vision-threating disease characterized by SRF accumulation, thus causing localized RD. Although it is usually self-limiting, CSCR may recur chronically with permanent visual acuity deficits. Around 80–90% of CSCR patients in the acute stage show spontaneous visual recovery after approximately 4–6 months without medication. In our study, the mean remission time in the placebo group (without drug treatment) was approximately 3.5 months, which is consistent with previous findings. There are many well-developed medical treatments for CSCR; however, they are associated with various serious side effects that are dangerous and even fatal for patients. In addition, a laser approach, such as argon laser photocoagulation (ALP), photodynamic therapy (PDT), or trans-pupillary thermotherapy (TTT), has been used for treating CSCR [27].

Nowadays, CSCR is well-studied and is associated with a type A behavior pattern which, in turn, is associated with physiological changes, including increased blood pressure and elevated serum cortisol and epinephrine levels. In addition, the concentrations of plasma epinephrine correlate with central macular thickness, macular edema and vision [28,29]. Therefore, this is the theoretical basis for choosing propranolol (Inderal^®^), a β-blocker typically used in the treatment of high blood pressure, to treat CSCR in our experimental design. The results showed that administering oral propranolol tablets shortened the disease course to a mean time of 1.9 months, and helped patients to promptly recover their BCVA. Furthermore, higher recurrence rates were observed even if patients had received treatment. For example, following spironolactone treatment, a higher recurrence rate of approximately 48% has been noted within 3–6 months after recovery [24]. Moreover, a 24.5% recurrence rate has also been observed after photodynamic therapy in subjects with chronic CSCR [26]. In our study, propranolol treatment resulted in a decrease in the recurrence rate to 5.3%, which is beneficial for CSCR patients.

There are many different types of medical treatment for CSCR, including the use of β-blockers. However, their various associated side effects have proven dangerous and even fatal for patients. Therefore, laser approaches have been recently developed as an alternative. For example, ALP, PDT or TTT should be used for CSCR subjects. However, such modern and invasive techniques involve complicated procedures and a higher cost, which causes hesitation in patients [30]. Furthermore, many cases of acute CSCR are not eligible or do not respond to treatment with TTT or PDT [18,31]. Because the complications of the various methods are serious and complex, many researchers have worked hard to try to develop easier, safer, and cheaper treatments for CSCR that are both rapid and efficient.

Because of the controversial role of steroids in CSCR treatment, several methods have been adopted in the past, such as those involving ketoconazole (inhibits the steps of glucocorticoid synthesis and decreases the level of cortisol), mifepristone (RU-486^®^, Lotus, Taiwan) (an antagonist of glucocorticoids and progesterone), carbonic anhydrase inhibitors (decrease excessive extracellular fluid), bevacizumab (Avastin^®^, Roche, Switzerland) (reduces serous RPE detachment and enhances absorption of SRF in the macula), and fenofibrate. As such, there are many medical treatments for CSCR patients; however, their various side effects are dangerous and even fatal for patients. Instead, a laser approach, for instance argon laser photocoagulation, PDT or TTT, should be used for CSCR subjects. However, such modern and invasive techniques involve complicated procedures and a higher cost, which causes hesitation in patients [30]. Furthermore, many cases of acute CSCR are not eligible or do not respond to treatment with TTT or PDT. Therefore, in our study, we strived to develop a new method for CSCR patients.

Recently, some ophthalmologists have made use of various β-blockers to treat CSCR, in which the possible mechanisms are related to the modification of choroidal circulation. It is well-known that the overexpression of receptors, specifically in the vascular endothelium, carries many features of acute CSCR [31]. Therefore, blocking the available β-receptors from the sympathetic and para-sympathetic systems and regulating choroidal blood flow may benefit subjects with CSCR. It has been reported that excessive epinephrine in ocular tissues affects the retina through β-adrenergic receptors on RPE cells. Therefore, the activation of receptors may produce changes in cyclic adenosine monophosphate concentration (cAMP), which, in turn, affects the electrical activity of RPE cells. Therefore, β-blockers could prevent the changes in RPE activity and epinephrine-induced apoptosis that compromise the integrity of RPE cells, and contribute to treating patients with CSCR. In recent years, some β-blockers (e.g., propranolol, nadolol, and metoprolol) have been used to try to treat CSCR subjects, although different results have been shown even for the same beta-receptors [32,33,34,35,36] (Table 3). Furthermore, we compared the past and associated reports from a few documents. Although there was a spontaneous improvement without treatment, β-blockers showed a plausible mechanism of action in CSCR. Chrapek and his co-workers published their study on the treatment of acute CSCR using trimepranol (metipranolol^®^, Alcon, Belgium) at a dose of 5 mg twice per day. The results showed rapid enhancement of the reattachment of the macula neuroepithelium within three months of therapy; in 8.8 weeks, on average, in the small group (only 13 eyes) [33]. Moreover, the success rate reached 84.6% (11/13) and the rate of failure was 15.4% (2/13). When comparing Chen’s report with our study, the use of propranolol may be shown to enhance the higher success rate and lower the recurrence rate. Indeed, trimepranol is a non-selective adrenergic receptor-blocking agent that does not have significant intrinsic sympathomimetic activity, and only has mild direct myocardial depressant activity [33]. Therefore, our study demonstrated a comparatively larger evidence-based study (total of 120 CSCR patients), and the results showed more reliable and exact conclusions in terms of the statistics. Moreover, the complete remission time was shorter in our research (1.9 months vs. 8.8 weeks). Charpek’s prospective double-blind study involved 48 eyes with the first attack of CSCR (not exceeding four weeks) who received 10 mg of trimepranolol twice daily; however, it was interesting to find that there was no effect of trimepranolol on the duration of CSCR when compared with a placebo group [36] (Table 3). Why could higher doses of trimepranolol not enhance the duration of this disease? It could be concluded that the cause of macula neuroepithelium detachment in CSCR is very complex, involving mechanisms not currently fully understood and a dysfunction that cannot be treated completely by trimepranol [36]. Another study included 21 patients with CSCR taking trimepranolol (a non-selective β-blocker) at a dose of 10 mg twice a day, and 30 CSCR subjects administered vasocardin (metoprolol^®^, AstraZeneca, Sweden) (a selective β-blocker) at a dose of 50 mg twice per day for several months. However, Fabianová et al. found that the cessation time and the time of relapse of CSCR were not significantly different between trimepranolol and vasocardin [21,32]. Hence their conclusion that the selective characteristics of β-receptors were not dominant in ocular tissues when treating CSCR patients. They even demonstrated that there was no difference in selective or non-selective beta-blockers. In our study, taking propranolol effectively decreased the recurrence rate (5.3%) in CSCR patients, remarkably more so than trimepranolol and vasocardin [32]. Therefore, we suggest that propranolol may be the more effective β-blocker agent in controlling the time course of CSCR. Moreover, nadolol is also a non-selective β-blocker and may inhibit the effects of catecholamine.

Browing and his colleagues used oral nadolol (40 mg daily) for eight CSCR patients, and found that it belongs to one of the types of beta-blockers that do not benefit CSCR subjects; all cases of treated patients resulted in failure [34], though the exact reasons and mechanisms need to be examined in detail in the future. Furthermore, in Tatham’s report, propranolol (40 mg twice per day) was administered for only two CSCR subjects [35]. Patient 1 experienced an increase in vision and a remission of other signs after taking propranolol; however, patient 2, a 28-year-old female, did not receive the oral beta-blocker therapy until the end of the second month. After nine months, her BCVA had returned to 6/6; however, symptom relief was only temporary, and her BCVA subsequently dropped to 6/9 and distortion persisted [35]. Therefore, taking propranolol would be the most appropriate and useful treatment protocol for treating CSCR. However, according to the above experience, the doses of propranolol should be prescribed for rapid recovery time and for subsequently preventing recurrence without hesitation [35]. Furthermore, Tatham et al. reported that the indicated doses of propranolol should be administered as quickly as possible to reduce the risk of recurrence [35]. To our surprise, the same prescribed dose of propranolol showed different outcomes in CSCR between Tatham’s research and ours. Comparing the above five studies with our experiment, we have summarized four conclusions (see Table 3). First, not all β-blockers may be effective for the clinical condition of CSCR. Second, the ability of propranolol to treat CSCR is stronger than that of other β-blockers. Third, in answer to why various types of β-blockers have different effects on CSCR patients, we suppose that the different clinical functions of various β-blockers may be due to their specific biochemical structures and properties, which deserve further evaluation. Additionally, the doses of β-blockers and patient body weight may also be key points to consider. Finally, the possibilities of pharmacokinetic theories for propranolol, and various pathways for different β-blockers, should be investigated in the future. Recently, the role of propranolol has become clearer in some of the steps of biometabolism. It can reduce platelet adhesion and aggregation, and block pathophysiological reactions caused by sympathetic nerve excitement [37,38,39]. Moreover, propranolol can reduce angiotensin-II levels, decrease the secretion of a variety of angiogenic factors, especially VEGF, inhibit the formation of single-layer endothelial cells, and counter the effects of catecholamine in the pathology and clinical condition of CSCR [40]. Finally, if deciding upon the method of oral Inderal^®^, this agent should be prescribed as soon as possible to avoid an increased possibility of recurrence. It is well-known that some β-blockers may also interact with receptors in bronchial and cardiovascular tissues, potentially resulting in sides effects [29].

According to the report of Harth, a single dose of beta-blocker does not reduce asthmatic symptoms. Furthermore, they concluded that cardioselective beta-blockers do not produce clinically significant adverse effects on respiratory function in patients with mild to moderate reactive airway disease and that, on the other hand, that cardioselective beta-blockers should not be withheld from patients with mild to moderate reactive airway diseases [29,41]. The results reveal that relatively lower doses are safe. However, treatment protocols require several months, and the cumulative doses of beta-blockers may pose a hazard for CSCR patients with asthma or cardiovascular diseases. Moreover, we analyzed the beta-blockers that have previously been used to treat CSCR patients (Table 3), and found that trimepranolol, vasocardin, nadolol and propranolol are hazardous to CSCR patients with asthma or cardiovascular disorders to some degree. By contrast, metipranolol does not attack respiratory and cardiogenic targets; however, this agent failed to treat CSCR [36]. To date, no safe beta-blockers have been found for CSCR patients with asthma and/or cardiovascular disorder. Therefore, it was worthwhile to develop new and safe beta-blockers for CSCR subjects with contraindications of both asthma and heart diseases. In the meantime, IVI avastin^®^ (Roche, Switzerland) or some type of medical laser could be alternative and acceptable choices.

In our larger evidence-based study, the success rate was approximately 95.0% (57/60), and the mean remission time of CSCR was shorter (1.9 months) than that in other studies and in spontaneous remission. We prescribed oral propranolol at doses according to the subject body weight (30–40 mg daily). Furthermore, we compared the amount of propranolol between Tatham’s design (80 mg daily) and ours (30 or 40 mg daily), and the lower doses from our protocol showed relatively higher safety for CSCR patients because they avoid possible (severe) complications such as bronchospasm or bradycardia. In other words, the regimen in our study was determined to be less harmful to CSCR patients with asthma and severe cardiovascular diseases. Moreover, no obvious personal discomfort was noted in the 120 patients during our study. Finally, we demonstrated that lower doses (30 or 40 mg/day) of propranolol for at least four months may facilitate SRF absorption, decrease the mean remission time (1.9 months), increase the success rate (95.0%), enhance the improvement of BCVA and diminish the recurrence rate (5.3%) compared to the protocols used in other published studies. Therefore, we propose that our new treatment method be considered as another choice for the treatment of patients with CSCR.

## 4. Materials and Methods

### 4.1. Patients and Study Design

From April to December 2017, we conducted a prospective study of 120 CSCR patients, aged between 20 and 50 years of age. The patients were randomly divided into groups 1 and 2, with 60 subjects in each group. The baseline characteristics of the two groups, including who received propranolol and who received the placebo, are listed in Table 2. The ratio between male and female was 50:10 and 47:13 in group 1 and group 2, respectively, revealing that CSCR indeed affects men more than women [3]. Signs of CSCR were detected by optic coherence tomography (OCT; OPKO, the E-Vision Instrument Company, Taiwan) scanning.

### 4.2. Patients and Criteria

All 120 patients were instructed to return to our clinics every week for a series of examinations, including for BCVA and an OCT scan. First, we recorded the initial BCVA at baseline when the study started, as well as information on the eyes (either left or right), age, sex, and body weight of each patient. Using OCT scans, we measured changes in the height of the dome shape that indicated the amount of SRF accumulation, which coincided with patients’ BCVA. An elevated dome shape in the OCT scans was the predominant sign of CSCR at the sub-RPE space around the macula. We evaluated the height of these dome shapes in the OCT images at 10:00 a.m. during each monthly follow-up. Because the elevated dome shape reveals SRF accumulation as a result of abnormal choroidal circulation, we are able to precisely confirm CSCR. If the image from an OCT scan appeared flat, this indicated the complete absorption of SRF and the “success” of this treatment for the patient. At the same time, the time from baseline to complete SRF absorption was defined as “the complete remission time” (Figure 2 and Figure 3). The dome shape can be found in Figure 2, highlighting subretinal fluid accumulation, which is the predominant sign of CSCR. After treatment by propranolol, the fluid was absorbed and the dome shape disappeared. This resulted in the attachment of the neurosensory retinal layer and improvement with respect to associated problems, as shown in Figure 3. Patients with severe cataracts, glaucoma, uveitis, ocular trauma, abnormal lesions in the retina or choroid, and S/P ocular surgery, or receiving any treating methods for CSCR, were excluded. For example, patients who had ever received treatment involving IVI of bevacizumab or steroids were excluded because of the possible interactions with propranolol [23]. Propranolol is a non-selective β-antagonist agent, and subjects with asthma or cardiovascular diseases had to be ruled out for safety reasons. Whenever the patients felt any discomfort (e.g., shortness of breath, chest pain, or bradycardia) at any time during the study, they were required to stop taking propranolol at once due to safety concerns.

### 4.3. Experimental Design

All 120 cases were randomly distributed and divided into two groups (60 CSCR patients in each group) according to our experimental design. In group l, 60 patients were enrolled, and oral propranolol was prescribed according to the protocols. For a body weight of less than 50 kg, 30 mg of oral propranolol daily was prescribed; for body weight over 50 kg, 40 mg of propranolol daily was prescribed. In group 2, the 60 volunteers received a placebo (100 mg vitamin C daily). Because some patients with CSCR have been known to show spontaneous remission at the borderline of 4 months without medication, all volunteers were required to take the propranolol or placebo for at least 4 months to avoid any bias and misunderstandings. In other words, for all of the “success” cases in groups 1 and 2, spontaneous recovery without medication should be ruled out.

When a patient was categorized as “successful” based on their OCT images after the 4-month treatment, the therapy was stopped for that patient, who then progressed to the 5-month follow-up program. The “successful” cases from groups 1 and 2 were followed for 5 months to evaluate the rate of recurrence after the different treatment methods. During the total 9 months, any side effects or uncomfortable sensations were monitored (4 months in treatment and 5 months in the follow-up phase). The BCVA of the CSCR patients was checked by the Snellen chart and, afterward, was translated to logMAR to enable a more accurate estimate of acuity in the research context. In the past, CSCR could only be diagnosed by fluorescein angiography (FAG), which identified the point-like hyperfluorescent findings in the early stage. After that, fluorescent dye would spread slightly from the leakage point over time, providing results equal to the results of neurosensory detachment in OCT images. Nowadays, OCT scans, instead of invasive FAG, are preferred for rapid diagnosis in the medical field. In the acute stage, OCT demonstrates RPE elevation or a pigment epithelium detachment (PED) occurrence at the leakage sites. Furthermore, in patients with chronic CSCR, there could be hyperreflective content over the Bruch’s membrane, creating a “double-layer sign” in OCT images. Therefore, we were able to easily analyze the characteristics of the “success” and “failure” of CSCR pictures from the OCT images after various therapies at monthly intervals.

### 4.4. Termination of Treatment

In our initial plan, all 120 CSCR patients were required to take the propranolol or placebo agents for 3 months. However, we found improved symptoms in some patients, including BCVA and color sensation, and the dome shape in the OCT scan was shown to be completely flat prior to the 3-month end date. Hence, the case could be classified as a “success” ahead of our planned time (4 months). At this point, the treatment was terminated and the volunteer began to receive follow-up checks over the next 5 months to evaluate the possibility of recurrence. Furthermore, the treatment was also terminated at once, for safety reasons, if the patients complained about dyspnea, chest tightness, muscle pain, anxiety or general weakness, or any physiological reactions that could have been caused by the beta-blocker.

### 4.5. Statistical Analysis

All of the results obtained in this study are expressed as mean ± standard deviation (SD). We assessed the normality of data distribution prior to using parametric statistical tests [3]. Moreover, the change in mean BCVA, the mean of total remission time, the success rate, and the recurrence rate for the evaluation of treatment outcome were analyzed using the chi-square test to compare the two groups. All data analysis was conducted using SPSS version 13.0 (Chicago, IL, USA). When *p* < 0.05, the differences were considered statistically significant.

## 5. Conclusions

CSCR is a potentially sight-threatening condition with a complex pathogenesis. Some cases of CSCR are able to resolve spontaneously after approximately four months without taking drugs. In this study, we identified the clinical parameters that would become the outcome of CSCR by our newly designed protocols. The aim of this study was to determine what the patients would benefit from when undergoing earlier therapeutic intervention by absorbing SRF. The treatment included propranolol, which may decrease the higher recurrence rate and could accelerate the recovery time, thus ameliorating patients’ concerns. We suggest that epinephrine in the pathophysiology of CSCR and β-blockade may play roles in treating the condition [32]. Herein, we conducted a large study of CSCR patients who were administered propranolol treatment, and our proposed method was demonstrated to be a better, faster, safer, and easier strategy for the treatment of CSCR patients.

## Figures and Tables

**Figure 1 pharmaceuticals-13-00336-f001:**
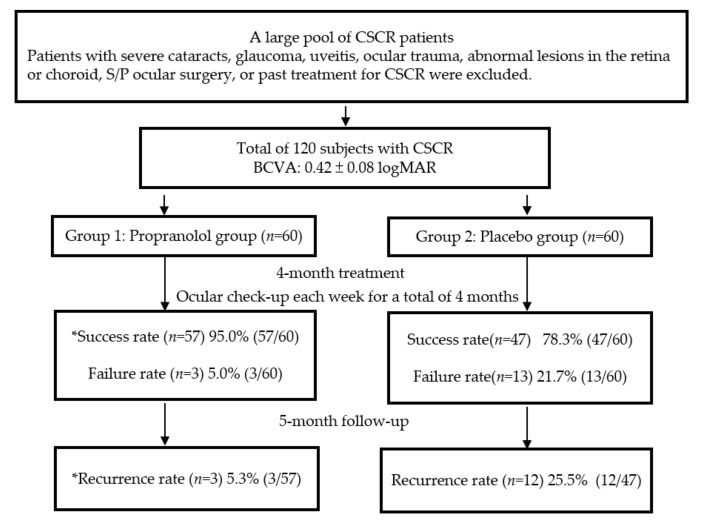
The flow chart shows the designed protocols and final results, divided into groups 1 and 2, during the whole 9 months of our study. * denoted a significant difference between the propranolol and placebo groups.

**Figure 2 pharmaceuticals-13-00336-f002:**
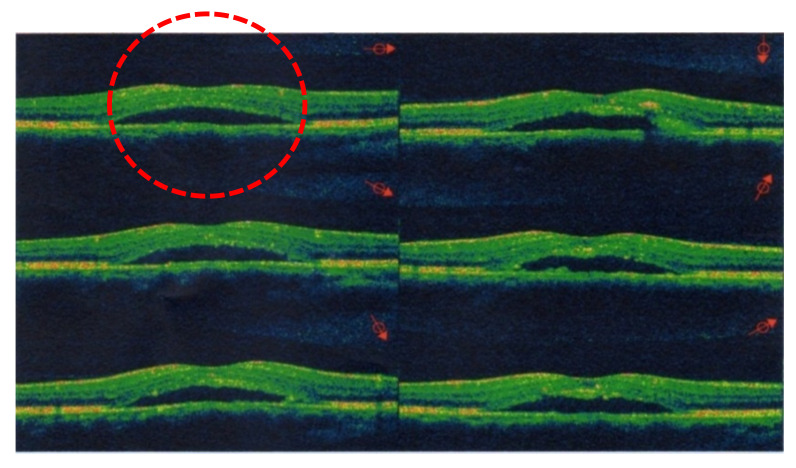
Optical coherence tomography (OCT) images generated using the OPKO (E-Vision Instrument Company, Taiwan) in a 42-year-old male with CSCR. The topographic maps show highly bulged swelling from macular retinal edema and loss of the fovea at day 1.

**Figure 3 pharmaceuticals-13-00336-f003:**
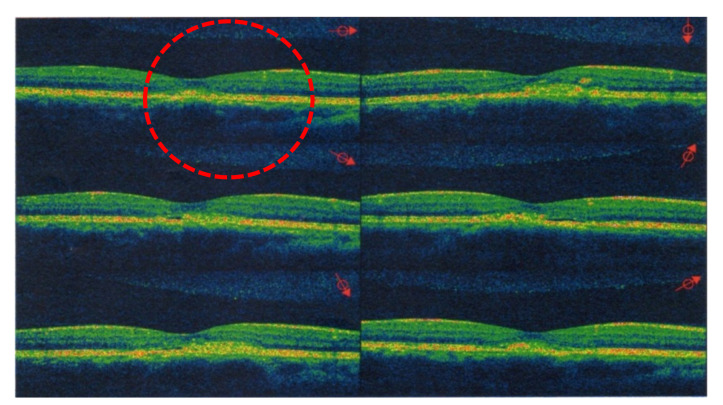
After oral propranolol for 70 days (Day 70), the vertical and horizontal scans from the 42-year-old male with CSCR show that the retinal pigment epithelium (RPE) had detached with significantly reduced macular edema. Therefore, the patient was considered as a “success” case for propranolol treatment in group 1.

**Table 1 pharmaceuticals-13-00336-t001:** Changes in the parameters after our treatment protocols in the 120 patients with CSCR.

	Group	Group 1 (*n* = 60) Propranolol Treatment	Group 2 (*n* = 60) Placebo Treatment	*p*-Value
Parameter				
Mean complete remission time		* 1.9 months	3.5 months	0.008
Success rate		* 95.0% (57/60)	* 78.3% (47/60)	0.001
Mean BCVA		* 0.09 ± 0.01 logMAR	0.19 ± 0.03 logMAR	0.032
Rate of recurrence		* 5.3% (3/57)	25.5% (12/47)	0.14

Notes: The length of effective oral propranolol treatment was only 1.9 months, which is shorter than that in most other documented research studies, with an excellent success rate (95.0%) and a diminishing of the original high recurrence rate (5.3%) observed for the 5-month follow-up period. The differences in all four parameters were associated with *p*-values of less than 0.05. Therefore, the difference between groups 1 and 2 was considered statistically significantly. * denoted a significant difference between the propranolol and placebo groups.

**Table 2 pharmaceuticals-13-00336-t002:** The baseline characteristics of the propranolol and placebo groups.

	Group	Group 1 (Propranolol Group)	Group 2 (Placebo Group)
Parameter			
Eyes involved		60 patients (60 eyes)	60 patients (60 eyes)
The drug for treatment		Propranolol	Vitamin C
The given dose		2 × 20 mg/day	100 mg/day
Mean age (years)		42.5 ± 2.6	43.8 ± 3.4
Male/female ratio		50:10	47:13

Notes: The patients with central serous chorioretinopathy (CSCR) received treatment by taking propranolol (2 × 20 mg/day) daily. In group 1, daily oral propranolol was prescribed according to the patient’s body weight. In group 2, all 60 CSCR subjects were required to ingest 100 mg of vitamin C daily. The ratio of males/females was 50:10 and 47:13 in group 1 and group 2, respectively, which reveals that males are more likely to develop CSCR.

**Table 3 pharmaceuticals-13-00336-t003:** A comparison of various beta-blockers in treating central serous chorioretinopathy.

	Parameters	Drugs	Doses	Eyes	Results and Outcomes
Research Group					
Fabianová et al. [18]		Trimepranol Vasocardin	2 × 10 mg/day 2 × 50 mg/day	21 30	1. The average remission time was 4.5–4.8 weeks. 2. No difference in selective or non-selective blockers.
Chrapek et al. [19]		Trimepranol	2 × 5 mg/day	13	1. Success rate: 84.6%. 2. Failure rate: 15.4%. 3. Complete remission time: 8.8 weeks. 4. Trimepranol was not reliable.
Browing [20]		Nadolol	40 mg/day	4	1. Failure rate: 100%. 2. Nadolol had adverse effects.
Tatham et al. [23]		Propranolol	2 × 40 mg/day	2	1. VA recovery and OCT became flat after 72 days. In addition, successful for both eyes. 2. Recurrence after 2 months likely.
Chrapek et al. [21]		Metipranolol	1 × 10 mg/day	23	1. No significant difference between metipranolol and placebo therapy. 2. No effect on CSCR.
Chen et al. (our study)		Propranolol	2 × 20 mg/day	60	1. Remission: 1.9 months. 2. Success rate: 95%. 3. Recurrence: 5.3%.

Notes: The trade name of trimepranolol is Trimepranol^®^ ( Falcon, USA), which is a type of non-selective β-blocker. The trade name of vasocardin is Vasocardin^®^ (Mylan, USA), which is a β-blocker agent. The trade name of propranolol is Inderal^®^ (AstraZeneca, United Kingdom), which belongs to the group of β2-adrenergic agonists and can induce an asthma attack. The trade name of nadolol is Corgurd l^®^ (Bristol-Myers Squibb, United Kingdom) or Nadolol l^®^ (Bristol-Myers Squibb, United Kingdom), which are non-selective beta-blockers. The trade name of metipranolol is Metoprolol succinate^®^ (AstraZeneca, Sweden) or Betaloc ZOK^®^ (AstraZeneca, Sweden).

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
