# Peer review of "Oral Treatment of Central Serous Chorioretinopathy Patients Using Propranolol Tablets"

_pharmaceuticals, 2020, doi:10.3390/ph13110336_

Round 1
Reviewer 1 Report
The manscript entitled” Treating Central Serous Chorioretinopathy Patients with Propranolol Oral Tablets” is very interesting and valuable. CSCR is a very common ocular disease in clinics. However, the treatments varied and the associated side effects from the drugs bring about damages to the patients. The experiment results showed that the propranolol have potential to treat CSCR patients, enhanced subretinal fluid (SRF) absorption, shorten the time of total complete remission and decrease CSCR recurrence.
This manuscript can be accepted after minor revision. I recommended:
1. In line 12, please correct “introduction” as “purpose”.
2. In line 17, please use the full English name of “BCVA” in the first time.
3. Please describe the onset of the CSCR of the victims in this selected criteria? Within 2 weeks or 4 weeks ? Acute CSCR or chronic CSCR ?
4. Inderal may be contraindication for patients with asthma and C-V disorders that may induce bradycardia or constriction of the trachea. Besides, even corticosteroid is good for asthma. However, long-term use of steroid may exacerbate the diseases course of CSCR. What treatment method you suggest if the patient have both CSCR and asthma?
5. Please highlight the differences in Fig1, and Fig 2.
6. Please revise the importance of this study in conclusion section.
Author Response
Dear reviewer,
Thank you for the precious suggestions on the manuscript, “Treating Central Serous Chorioretinopathy Patients with Propranolol Oral Tablets,” and we have made substantial revisions to the manuscript based on the reviewer’ comments, as the details below.
- In line 12, please correct “introduction” as “purpose”.
Response: We have replaced “purpose” with “introduction” in line 12.
In line 17, please use the full English name of “BCVA” in the first time.
Response: We have used the full name of best-corrected visual acuity (BCVA) in line 14.
- Please describe the onset of the CSCR of the victims in this selected criteria? Within 2 weeks or 4 weeks? Acute CSCR or chronic CSCR?
Response: We selected total 120 subjects who suffered from CSCR within 1 months. The onset of the CSCR time was less than 4 weeks, and it may be concluded as acute CSCR.
- Inderal may be contraindication for patients with asthma and C-V disorders that may induce bradycardia or constriction of the trachea. Besides, even corticosteroid is good for asthma. However, long-term use of steroid may exacerbate the diseases course of CSCR. What treatment method you suggest if the patient have both CSCR and asthma?
Response:
We have suggested the treatment method in lines 509-513.
“To date, no safe beta-blockers have been found for CSCR patients with asthma and/or cardiovascular disorder. Therefore, it was worthwhile to develop new and safe beta-blockers for CSCR subjects with contraindications of both asthma and heart diseases. In the meantime, IVI avastin® or some type of medical laser could be alternative and acceptable choices.”
- Please highlight the differences in Fig1, and Fig 2.
Response: We have marked the differences in Fig. 1 and Fig. 2.
“An elevated dome shape in the OCT scans was the predominant sign of CSCR at the sub-RPE space around the macula. We evaluated the height of these dome shapes in the OCT images at 10:00 a.m. during each monthly follow-up. Because the elevated dome shape reveals SRF accumulation as a result of abnormal choroidal circulation, we are able to precisely confirm CSCR. If the image from an OCT scan appeared flat, this indicates the complete absorption of SRF and “success” of this treatment for the patient. At the same time, the time from baseline to complete SRF absorption was defined as “the complete remission time” (Figs. 1 and 2). The dome shape can be found in Fig. 1, highlighting subretinal fluid accumulation, which is the predominant sign in CSCR. After treatment by propranolol, the fluid was absorbed and the dome shape disappeared. This resulted in the attachment of the neurosensory retinal layer and improvement with respect to associated problems, as shown in Fig. 2.” in lines 123-135.
- Please revise the importance of this study in conclusion section.
Response: We have revised the conclusion carefully. Besides, we also have described some importance concepts in the of conclusion section.
“CSCR is a potentially sight-threatening condition with a complex pathogenesis. Some cases of CSCR are able to resolve spontaneously after approximately four months without taking drugs. In this study, we identified the clinical parameters that would become the outcome of CSCR by our newly designed protocols. The aim of this study was to determine what the patients would benefit from when undergoing earlier therapeutic intervention by absorbing SRF. The treatment included propranolol, which may decrease the higher recurrence rate and could accelerate the recovery time, thus ameliorating patients’ concerns. We suggest that the evidence for epinephrine in the pathophysiology of CSCR andβ-blockade may play roles in treating the condition [34]. Herein, we conducted a large study of CSCR patients who were administered propranolol treatment and our proposed method was demonstrated to be a better, faster, safer, and easier strategy for treatment of CSCR patients.” in lines 532-543.
Kind regards
With above responses and substantial revisions, please kindly consider its publication in Pharmaceuticals. Thank you for your kind consideration and assistance.
Sincerely,
Chi-Ting Horng M.D., Ph.D.
Mail: h56041@gmail.com
Reviewer 2 Report
- Introduction did not cover the rationale for using beta blockers in CSCR.
- Please provide baseline characteristics for groups 1 and 2.
- Please provide more details on the placebo given to the control group.
- Did authors assess the normality of data distribution prior to using parametric statistical tests?
- Success rate and rate of recurrence are categorical variables. How did authors use Student t test to compare between groups?
- Please provide the exact p value rather than <0.05.
- What was William’s test used for in this study?
- References need to be added to many statements in the discussion.
Author Response
Dear reviewer,
Thank you for the precious suggestions on the manuscript, “Treating Central Serous Chorioretinopathy Patients with Propranolol Oral Tablets,” and we have made substantial revisions to the manuscript based on the reviewer’ comments, as the details below.
1. Introduction did not cover the rationale for using beta blockers in CSCR.
Response: We have modified and added the mechanism of beta blockers for CSCR in lines 76-94.
“Most medical reports suggest that without aggressive treatment, RPE atrophy and CNV can worsen in cases of CSCR [11-13]. Regardless of the characteristics of spontaneous remission in CSCR, the trend is to treat CSCR rather than wait for possible recovery by itself. Therefore, early prevention, diagnosis, and treatment are necessary, which could possibly enable complete remission of SRF and rapid recovery of vision within the shortest possible time [14]. Therefore, blocking the progression of CSCR and enhancing the absorption of SRF has quickly become mainstay in CSCR treatment. Thus far, there are several treatments that have been developed for CSCR patients. For example, several years ago, steroids were initially administered to CSCR patients either orally or intravenously [15]. Since then, various therapies have been developed, including the popular approach of intravitreous injection (IVI) of bevacizumab (Avastin®) [16]. Moreover, Gramajo et al. reported that melatonin (3 g three times daily) may decrease the central thickness of macula and improve BCVA in chronic CSCR [17]. However, an aggressive approach has also been adopted for CSCR patients and includes thermal laser photocoagulation and photodynamic therapy, which may carry inherent risks and show poor results [18,19]. In this research, we attempted to develop a new method for treating patients with CSCR based on only taking propranolol. Beta-blockade is useful in clinical conditions in which sympathetic activities are excessive—in this case, the pathophysiology of CSCR—which may impact on visual function [20]. Furthermore, anxiety that increases the catecholamine level may also be a factor impacting CSCR. Therefore, we supposed that the β-blockers, such as propanol that was used in this study, are potential drugs for curing CSCR [21].”
2. Please provide baseline characteristics for groups 1 and 2
Response: We have added the table 1 and description in 2.3 Experimental design section in lines 128-142 .
“All 120 cases were randomly distributed and divided into two groups (60 CSCR patients in each group) according to our experimental design. In group l, 60 patients were enrolled, and oral propranolol was prescribed according to the protocols. For a body weight of less than 50 kg, 30 mg of oral propranolol daily was prescribed; for body weight over 50 kg, 40 mg of propranolol daily was prescribed. In group 2, the 60 volunteers received a placebo (100 mg vitamin C daily). Because some patients with CSCR have been known to show spontaneous remission at the borderline of 4 months without medication, all volunteers were required to take the propranolol or placebo for at least 4 months to avoid any bias and misunderstandings. In other words, for all of the “success” cases in groups 1 and 2, spontaneous recovery without medication should be ruled out”.
Table 1. The baseline characteristics of the propranolol and placebo groups.
|
Group |
Group 1 (propranolol group) |
Group 2 (placebo group) |
|
Eyes involved |
60 patients (60 eyes) |
60 patients (60 eyes) |
|
The drug for treatment |
Propranolol |
Vitamin C |
|
The given dose |
2 × 20 mg/day
|
100 mg/day
|
|
Mean age (years)
|
42.5 ± 2.6
|
43.8 ± 3.4
|
|
Male/female ratio |
50:10 |
47:13 |
Notes: The patients with central serous chorioretinopathy (CSCR) received treatment by taking propranolol (2 × 20 mg/day) daily. In group 1, daily oral propranolol was prescribed according to the patient’s body weight. In group 2, all 60 CSCR subjects were required to ingest 100 mg of vitamin C daily. The ratio of males/females was 50:10 and 47:13 in group 1 and group 2, respectively, which reveals that males are more likely to develop CSCR.
3. Please provide more details on the placebo given to the control group.
Response: The details on the placebo given to the control group have added in lines 131-136.
“In group 2, the 60 volunteers received a placebo (100 mg vitamin C daily). Because some patients with CSCR have been known to show spontaneous remission at the borderline of 4 months without medication, all volunteers were required to take the propranolol or placebo for at least 4 months to avoid any bias and misunderstandings. In other words, for all of the “success” cases in groups 1 and 2, spontaneous recovery without medication should be ruled out.”
4. Did authors assess the normality of data distribution prior to using parametric statistical tests?
Response: We have assessed the normality of data distribution prior to using parametric statistical tests. We also referred to and added the reference [30] in this study.
5. Success rate and rate of recurrence are categorical variables. How did authors use Student t test to compare between groups?
Response: According to the reviewer’s comments, we have used the Chi-square test (the sample number is greater than 60) to compare the categorical variables of the success rate, failure rate and the rate of recurrence in Fig. 3.
“We also compared the results between groups 1 and 2 using a chi-square test. A p-value <0.05 was considered statistically significant and is marked with an asterisk.” in lines 243-246.
6. Please provide the exact p value rather than <0.05.
Response: We have provided the exact p value in table 2.
Table 2. Changes in the parameters after our treatment protocols in the 120 patients with CSCR.
|
Group |
Group 1 (n = 60) propranolol treatment |
Group 2 (n = 60) placebo treatment |
p-Value |
|
Mean complete remission time |
* 1.9 months |
3.5 months |
0.008 |
|
Success rate |
* 95.0% (57/60) |
* 78.3% (47/60) |
0.001 |
|
Mean BCVA |
* 0.09 ± 0.01 logMAR |
0.19± 0.03 logMAR |
0.032 |
|
Rate of recurrence |
* 5.3% (3/57) |
25.5% (12/47) |
0.14 |
Notes: The length of effective oral propranolol treatment was only 1.9 months, which is shorter than that in most other documented research studies, with an excellent success rate (95.0%) and diminishing of the original high recurrence rate (5.3%) observed for the 5-month follow-up period. The differences in all four parameters were associated with p-values of less than 0.05. Therefore, the difference between groups 1 and 2 was considered statistically significantly. In our study, William’s test was used to compare the minimum difference (success rate) between the two groups (i.e., treatment and placebo) at the end of the study (the criterion was that the group size was greater than 60). Therefore, we have marked the success rate in the group 1 (treatment group) with #. BCVA, best-corrected visual acuity.
7. What was William’s test used for in this study?
Response: “The William’s test was used to compare the minimum difference in the success rate between the two groups at the end of the study. Therefore, we have marked the success rate in group 1 (treatment group) with #.” in lines 203-205.
8. References need to be added to many statements in the discussion.
Response: We have modified and added references in the discussion section.
Kind regards
With above responses and substantial revisions, please kindly consider its publication in Pharmaceuticals. Thank you for your kind consideration and assistance.
Sincerely,
Chi-Ting Horng M.D., Ph.D.
Mail: h56041@gmail.com
Reviewer 3 Report
The study by Chen et al evaluated the effect of oral propranolol in 120 patients with central serous chorioretinopathy (CSR). This study focuses on an important retinal disease that is commonly seen in clinical practice. The data suggest that oral propranolol 40 mg/day can enhance recovery as well as decrease recurrences for CSR.
Major concerns
1) Overall, the manuscript warrants extensive editing in terms of language, grammar, sentence structure, terminology and typo errors.
2) Several past studies have examined the effect of oral propranolol in patients with CSR. In what ways this study is different from the existing studies in the literature?
Minor concerns
- Page 1, Introduction section is overall too lengthy…suggest considerable shortening.
- Page 2, first line ‘middle-aged men almost between 45 and 55’: Please check. Generally CSR presents in relatively younger age. In fact, your own study shows mean age as 35.5 years.
- Page 2, second full paragraph, first sentence starting with ‘most medical reports..’ please reference this sentence.
- Page 2, second full paragraph, third and fourth last sentence describing treatment of CSR: Please include thermal laser photocoagulation and photodynamic therapy as well.
- Page 3, Methods section, 2.3 experimental design: why did authors mention the study as randomized controlled trial? Please clarify.
- Page 3, Methods section, 2.4 termination of treatment, second sentence: what do authors mean by ‘inverted U shape from OCT’?
- Page 6, first paragraph: fourth, third and second last sentences should be deleted since the information contained in these sentences does not describe results.
- Discussion section is not well organized. May be authors should focus on discussing the ‘response to treatment’ first followed by ‘rate of recurrence’. Also, Page 7 lacks paragraphs and therefore appears monotonous and difficult to read.
- Page 6, Discussion section, first paragraph: who prescribed the propranolol tablets? Ophthalmologists or physicians?
- Page 6, discussion section, second paragraph: initial sentences before the last sentence that spans page 6 and 7 are not relevant and should be deleted. In other words, this paragraph should start with the sentence ‘recently, for example, some ophthalmologist..’
Author Response
Dear reviewer,
Thank you for the precious suggestions on the manuscript, “Treating Central Serous Chorioretinopathy Patients with Propranolol Oral Tablets,” and we have made substantial revisions to the manuscript based on the reviewer’ comments, as the details below.
Major concerns
1. Overall, the manuscript warrants extensive editing in terms of language, grammar, sentence structure, terminology and typo errors.
Response: The manuscript has been carefully revised by English Editing Services from MDPI. All the modifications of language in the manuscript have been marked by using the "Track Changes" function.
2. Several past studies have examined the effect of oral propranolol in patients with CSR. In what ways this study is different from the existing studies in the literature?
Response: We have added the table 3. (A comparison of various beta-blockers in treating central serous chorioretinopathy.) in lines 349-355, this made it easier for readers to understand the differences and the importance of this article.
Minor concerns
1.Page 1, Introduction section is overall too lengthy…suggest considerable shortening.
Response: We have shortened and modified the introduction section
2. Page 2, first line ‘middle-aged men almost between 45 and 55’: Please check. Generally CSR presents in relatively younger age. In fact, your own study shows mean age as 35.5 years.
Response: The mean age of the 120 subjects was aged 45.5 ± 2.5 and has been corrected in line 227.
3. Page 2, second full paragraph, first sentence starting with ‘most medical reports..’ please reference this sentence.
Response: We have added the references [11-13] in line 83.
- Daruich A, Matet A, Marchionni L, et al. Acute central serous chorioretinopathy. Retina 2017; 37(10): 1905-15.
- Maruko I, Iida T, Ojima A, et al. Subretinal dot-like precipitates and yellow materials in central serous chorioretinopathy. Retina 2011; 31(4): 759-65.
- Bousquet E, Beydoun T, Zhao M, et al. Mineralcortocoid receptor antagonism in the treatment of chronic central serous chorioretinopathy: a pilot study. Retina 2013; 33: 2096-2102.
4. Page 2, second full paragraph, third and fourth last sentence describing treatment of CSR: Please include thermal laser photocoagulation and photodynamic therapy as well.
Response: We have added the thermal laser photocoagulation and photodynamic therapy for treatment of CSCR in lines 94-104.
“However, an aggressive approach has also been adopted for CSCR patients and includes thermal laser photocoagulation and photodynamic therapy, which may carry inherent risks and show poor results [18,19]. In this research, we attempted to develop a new method for treating patients with CSCR based on only taking propranolol. Beta-blockade is useful in clinical conditions in which sympathetic activities are excessive in this case, the pathophysiology of CSCR—which may impact on visual function [20]. Furthermore, anxiety that increases the catecholamine level may also be a factor impacting CSCR. Therefore, we supposed that the β-blockers, such as propanol that was used in this study, are potential drugs for curing CSCR [21].”
5. Page 3, Methods section, 2.3 experimental design: why did authors mention the study as randomized controlled trial? Please clarify.
Response: “All 120 cases were randomly distributed and divided into two groups (60 CSCR patients in each group) according to our experimental design. In group l, 60 patients were enrolled, and oral propranolol was prescribed according to the protocols. “ in lines 146-149.
6. Page 3, Methods section, 2.4 termination of treatment, second sentence: what do authors mean by ‘inverted U shape from OCT’?
Response: We have corrected “inverted U shape” to “dome shape” in line 183-187.
“In our initial plan, all 120 CSCR patients were required to take the propranolol or placebo agents for 3 months. However, we found improved symptoms in some patients, including BCVA and color sensation, and the dome shape in the OCT scan was shown to be completely flat prior to the 3-month end date.”
7. Page 6, first paragraph: fourth, third and second last sentences should be deleted since the information contained in these sentences does not describe results.
Response: We have deleted these sentences according to the reviewer’s comments.
8. Discussion section is not well organized. May be authors should focus on discussing the ‘response to treatment’ first followed by ‘rate of recurrence’. Also, Page 7 lacks paragraphs and therefore appears monotonous and difficult to read.
Response: We have modified the discussion section. We have also focused on the response to treatment, and rate of recurrence. The flow chart showed that the designed protocols and final results includnig group1 and group 2 during the whole 9 months in this study as shown in Fig. 3.
9. Page 6, Discussion section, first paragraph: who prescribed the propranolol tablets? Ophthalmologists or physicians?
Response: “In our design, oral propranolol (Inderal®) tablets were prescribed by the same ophthalmologist (Dr. Horng) and used to treat the patients with CSCR.” in lines 327-329.
10. Page 6, discussion section, second paragraph: initial sentences before the last sentence that spans page 6 and 7 are not relevant and should be deleted. In other words, this paragraph should start with the sentence ‘recently, for example, some ophthalmologist..’
Response: We have deleted lines 334-335, and this paragraph have been started with the sentence’ for example’ in line 337.
Kind regards
With above responses and substantial revisions, please kindly consider its publication in Pharmaceuticals. Thank you for your kind consideration and assistance.
Sincerely,
Chi-Ting Horng M.D., Ph.D.
Mail: h56041@gmail.com
This manuscript is a resubmission of an earlier submission. The following is a list of the peer review reports and author responses from that submission.